



# Version 2 of the global catalogue of large anthropogenic and volcanic SO₂ sources and emissions derived from satellite measurements

Vitali E. Fioletov[1], Chris A. McLinden[1], Debora Griffin[1], Ihab Abboud[1], Nickolay Krotkov[2], Peter J. T. Leonard[3], Can Li[2,4], Joanna Joiner[2], Nicolas Theys[5], and Simon Carn[6]

[1]Air Quality Research Division, Environment and Climate Change Canada, Toronto, ON, Canada
[2]Atmospheric Chemistry and Dynamics Laboratory, NASA Goddard Space Flight Center, Greenbelt, MD, USA
[3]The Terrestrial Information Systems Laboratory, NASA Goddard Space Flight Center, Greenbelt, MD, USA
[4]Earth System Science Interdisciplinary Center, University of Maryland, College Park, MD, USA
[5]Royal Belgian Institute for Space Aeronomy (BIRA-IASB), Brussels, Belgium
[6]Department of Geological and Mining Engineering and Sciences, Michigan Technological University, Houghton, MI 49931, USA

Correspondence to: Vitali.Fioletov@outlook.com or Vitali.Fioletov@ec.gc.ca

**Abstract.** Sulfur dioxide (SO₂) measurements from the Ozone Monitoring Instrument (OMI), Ozone Mapping and Profiler Suite (OMPS), and TROPOspheric Monitoring Instrument (TROPOMI) satellite spectrometers were used to update and extend the previously developed global catalogue of large SO₂ emission sources. This version 2 of the global catalogue covers the period of 2005-2021 and includes a total of 759 continuously emitting point sources releasing from about 10 kt yr⁻¹ to more than 4000 kt yr⁻¹ of SO₂ that have been identified and grouped by country and primary source origin: volcanoes (106 sources); power plants (477); smelters (74); and sources related to the oil and gas industry (102). There are several major improvements compared to the original catalogue: it combines emissions estimates from three satellite instruments instead of just OMI, uses a new version 2 of the OMI and OMPS SO₂ dataset, and updated consistent site-specific air mass factors (AMF) are used to calculate SO₂ vertical column densities (VCDs). The newest TROPOMI SO₂ data processed with the Covariance-Based Retrieval Algorithm (COBRA), used in the catalogue, can detect sources with emissions as low as 8 kt yr⁻¹ (in 2018-2021) compared to the 30 kt yr⁻¹ limit for OMI. In general, there is an overall agreement within ±12% in total emissions estimated from the three satellite instruments for large regions. For individual emission sources, the spread is larger: the annual emissions estimated from OMI and TROPOMI agree within ±13% in 50% of cases and within ±28% in 90% of cases. The version 2 catalogue emissions were calculated as a weighted average of emission estimates from the three satellite instruments using an inverse-variance weighting method. OMI, OMPS, and TROPOMI data contribute 7%, 5%, and 88% to the average respectively for small (< 30 kt y⁻¹) sources and 33%, 20%, and 47% respectively for large (> 300 kt y⁻¹) sources. The catalogue data show an approximate 50% decline in global SO₂ emissions between 2005 and 2021, although emissions were relatively stable during the last 3 years. The version 2 of the global catalogue has been posted at the NASA Global SO₂ Monitoring web site (Fioletov et al., 2022).



## 1        Introduction

Sulfur dioxide ($SO_2$) plays an important role in atmospheric processes that impact the environment, health, atmospheric
chemistry and climate (Hansell and Oppenheimer, 2010; Robock, 2000). It poses a direct hazard to public health (Longo et al.,
2010; Pope and Dockery, 2006) and, therefore, is a designated criteria air pollutant in many countries. $SO_2$ also leads to acid
deposition that affects terrestrial ecosystems (Dentener et al., 2006; Hutchinson and Whitby, 1977; Vet et al., 2014). Coal-
burning power plants, oil refineries and smelters are the primary anthropogenic emitters of $SO_2$ (Klimont et al., 2013; Smith
et al., 2011), while volcanoes are the primary natural source of $SO_2$ (Carn et al., 2017; Oppenheimer et al., 2011).

Due to strong absorption of UV radiation, it is possible to retrieve $SO_2$ vertical column density (VCD) from satellite
measurements in the UV part of the spectrum. Such retrievals were first performed using measurements by the Total Ozone
Mapping Spectrometer (TOMS) and the Solar Backscattered Ultraviolet (SBUV) instruments on Nimbus 7 satellite after a
large injection of volcanic $SO_2$ from the El Chichón eruption in 1982 (Krueger, 1983; McPeters et al., 1984). Industrial
emission sources were first detected from space using measurements by the Global Ozone Monitoring Experiment (GOME)
on the European Remote Sensing satellite 2 (ERS-2) (Eisinger and Burrows, 1998; Khokhar et al., 2008). Measurements by
the two subsequent satellite instruments, the SCanning Imaging Absorption spectroMeter for Atmospheric CHartographY
(SCIAMACHY), 2002-2012, on the ENVISAT satellite (Bovensmann et al., 1999) and the Global Ozone Monitoring
Experiment-2 (GOME 2) instrument, 2006-present, on MetOp-A (Callies et al., 2000) were used to detect and monitor
emissions from a few dozen sources (Fioletov et al., 2013). A new era of satellite $SO_2$ measurements started with the launch
of the Dutch–Finnish Ozone Monitoring Instrument (OMI) (Levelt et al., 2018, 2006) on NASA's Earth Observing System
(EOS) – Chemistry Aura spacecraft (Schoeberl et al., 2006) in 2004.  At that time, OMI had the highest spatial resolution (up
to $13 \times 24$ km²) of any UV satellite instrument and was able to provide daily, nearly global maps of $SO_2$ VCDs,  permitting
the analysis of long-term trends in $SO_2$ emissions on a regional and global scale (Krotkov et al., 2016).

        A catalogue of large $SO_2$ sources and their emissions estimated from OMI measurements was introduced six years
ago (Fioletov et al., 2016; McLinden et al., 2016). At that time, the catalogue included 491 continuously emitting point sources,
of which 76 were volcanoes, 297 were powerplants, 53 were smelters and 65 were sources related to the oil and gas industry.
The catalogue was updated annually, and additional sources were added with the most recent version of the catalogue including
588 sources and available from the NASA Global $SO_2$ Monitoring web site at https://so2.gsfc.nasa.gov/measures.html (last
access: 7 July 2022). The catalogue was used to update and improve available bottom-up emissions inventories used in air
quality and climate models (Liu et al., 2018; Ukhov et al., 2020), to evaluate the efficiency of industrial clean technology
solutions in reducing air pollution (Ialongo et al., 2018; McLinden et al., 2020), and to monitor changes in $SO_2$ emissions on
a large scale (Li et al., 2017). The catalogue estimates for volcanic sources were used to analyse volcanic $SO_2$ emissions (Carn
et al., 2017) and, using $SO_2$ as a proxy, to estimate volcanic carbon dioxide ($CO_2$) fluxes (Fischer et al., 2019). The approach
to catalogue large emission point sources was later applied to satellite measurements of ammonia ($NH_3$) (Van Damme et al.,
2018; Dammers et al., 2019) and nitrogen dioxide ($NO_2$) (Beirle et al., 2021).



There have been several important developments since publication of the original SO$_2$ catalogue. Satellite SO$_2$ measurements by the Ozone Mapping and Profiler Suite (OMPS) (Zhang et al., 2017) on the NASA–NOAA Suomi National Polar-orbiting Partnership (SNPP) spacecraft (Flynn et al., 2014; Seftor et al., 2014) and by the TROPOspheric Monitoring Instrument (TROPOMI) (Theys et al., 2017) on the ESA Copernicus Sentinel-5 Precursor (S-5P) spacecraft (Veefkind et al.,
2012) became available starting in 2012 and 2018, respectively. Their measurements are suitable for SO$_2$ emissions estimates and can provide additional inputs for the SO$_2$ catalogue (Fioletov et al., 2020; Theys et al., 2021; Zhang et al., 2017). A newer European Centre for Medium-Range Weather Forecasts (ECMWF) Reanalysis v5 (ERA5)  reanalysis version (C3S, 2017) provides wind data with a higher spatial and temporal resolution than the ERA-Interim reanalysis (Dee et al., 2011) used in the original SO$_2$ catalogue. A new version 2.0 of the SO$_2$ retrieval algorithm was developed for OMI and OMPS SO$_2$ retrievals
(Li et al., 2020c), and the entire OMI and OMPS data records were reprocessed in 2020-2021.

After the release of the version 2.0 OMI SO$_2$ product in 2020, production of the previous version 1.2 used in the original catalogue was discontinued, so it was necessary to recalculate the SO$_2$ emissions using version 2.0 and that was done in this study. There were also some additional improvements in this updated version of the catalogue, and we decided to call it version 2. The original catalogue was based on OMI SO$_2$ data that have a source detection limit of 30-40 kt yr$^{-1}$ (Fioletov et
al., 2015, 2016),  the whereas newest TROPOMI SO$_2$ data set processed with the Covariance-Based Retrieval Algorithm (COBRA) can detect sources with emissions as low as 8 kt yr$^{-1}$ (Theys et al., 2021) and therefore more sources can be "seen". In this study, we discuss the implications of the change in OMI SO$_2$ data product versions as well as further changes introduced in the catalogue. Note that the SO$_2$ emission estimation algorithm used here is identical to that in the original study (Fioletov et al., 2016) to assure the continuity of the old and new emissions estimates. For this reason, we do not use a newer version of
the emission estimation algorithm (Fioletov et al., 2017; McLinden et al., 2020) that can better handle multiple emission sources in close proximity, which we plan to utilize in subsequent versions of the catalogue.

This article introduces a new version 2 of the global catalogue of large SO$_2$ sources and their emissions. It is organized as follows: the data sets and the emission calculation algorithm are described in **Section 2**. **Section 3** discusses the differences between emissions estimated from different versions of the OMI algorithm and SO$_2$ emissions estimates from different satellite
instruments. An overview of the estimated emissions is given in **Section 4**. **Section 5** concludes the study.

## 2      Data sets

VCDs measured by three hyperspectral "push broom" UV satellite sensors: OMI, OMPS, and TROPOMI were used in this study. SO$_2$ VCDs are given as in Dobson Units (DU, 1 DU = 2.69•10$^{16}$ molec•cm$^{-2}$) and the estimated annual emissions are in metric kilotonnes of SO$_2$ per year (kt y$^{-1}$).



## 2.1 OMI and OMPS data

OMI was launched on NASA's EOS Aura satellite on 15 July 2004 (Schoeberl et al., 2006). Aura is in a sun-synchronous polar orbit and crosses the Equator at about 13:45 local time. OMI is a nadir-viewing UV–visible spectrometer that initially provided daily global coverage with a resolution of up to 13 km × 24 km at nadir (de Graaf et al., 2016). The OMI detector has 60 cross-track positions, however about half of its pixels have been affected by a field-of-view blockage and stray light (the so-called "row anomaly") after 2007 (Levelt et al., 2018). As in the original catalogue, the first 10 and last 10 cross-track positions were excluded from the analysis to limit the across-track pixel width from 24 km to about 40 km.

The OMPS Nadir Mapper, a UV spectrometer on board the NASA-NOAA Suomi NPP satellite, was launched in October 2011. The OMPS detector has 36 cross-track positions and a nadir resolution of 50 km × 50 km. Similar to the OMI data analysis, the first 2 and last 2 OMPS cross-track positions were excluded. Suomi NPP is also in a polar orbit and crosses the Equator at about the same time as Aura - at about 13:30 local time.

The original catalogue was based on the OMI $SO_2$ VCD data product calculated using algorithm version 1.2 that is based on principal component analysis (PCA) of OMI-measured radiances (Li et al., 2013). In this study, we used the version 2 OMI and OMPS PCA $SO_2$ data (Li et al., 2020). In the version 2, for each scene there are 6 different estimates of the $SO_2$ VCD in DU obtained by making different assumptions about the vertical distribution of $SO_2$. Users interested in anthropogenic $SO_2$ pollution are advised to pick VCDs produced by spectral fitting using $SO_2$ Jacobians that more accurately account for the effects of sun-satellite geometry, clouds, $O_3$, and surface reflectivity on OMI (and OMPS) sensitivity as well as use updated a priori $SO_2$ vertical profiles from chemical transport model (CTM) simulations (i.e., the ColumnAmountSO2 field in the OMI and OMPS datasets (Li et al., 2020a, 2020b)). In addition, version 2 also provides an estimate of the slant column density (SCD) produced by spectral fitting using $SO_2$ cross sections (i.e., epy SlantColumnAmountSO2 field). When converted to VCD using a site-specific air mass factor ($AMF_{site}$=SCD/VCD) (McLinden et al., 2014), this dataset can be used as a continuity product for the previous version . The differences between the emissions estimates using the two approaches are discussed in Section 3.

OMI data for the period 2005-2021 and OMPS data for the period 2012-2021 were analysed. Only Level-2 clear-sky data, defined as having a cloud radiance fraction of less than 0.3, were used for the catalogue. Measurements at solar zenith angles (SZA) more than 70° as well as measurements taken over snow or ice were excluded. As in the original catalogue, the retrieved $SO_2$ VCDs correspond to 1-km thick plumes located near the surface as we focus on anthropogenic and passive volcanic degassing sources. Typical standard deviations of the individual OMI VCDs over background areas are between 0.6 DU in the tropics and 1 DU at high latitudes. The same values for OMPS are 0.3–0.4 DU (Fioletov et al., 2020).

## 2.2 TROPOMI data

TROPOMI was launched on the ESA Copernicus S5-P satellite on October 13, 2017 (Veefkind et al., 2012). The instrument consists of UV-VIS-NIR spectrometers and $SO_2$ can be retrieved from the UV part of the measured spectra. The TROPOMI



detector has 450 cross-track positions, however the first and last 20 of them were excluded from our analysis due to a relatively high noise level (Fioletov et al., 2020). The spatial resolution for the centre of the swath was originally 3.5 km × 7 km (along track) and it was reduced to 3.5 km × 5.5 km after August 6, 2019.

The original operational TROPOMI differential optical absorption spectroscopy (DOAS)-based algorithm (Theys et al., 2017) produced $SO_2$ data with high spatial resolution that made it possible to study $SO_2$ emission sources in greater detail and detect sources that previously were below the sensitivity limits of OMI and OMPS (Fioletov et al., 2020). However, the noise level was relatively high and the data had large-scale variable biases (Fioletov et al., 2020; Theys et al., 2021). These issues were largely resolved by a new Covariance-Based Retrieval Algorithm (COBRA) (Theys et al., 2021). Moreover,

COBRA even demonstrated lower uncertainties than the PCA-based algorithm when the same set of samples was processed by the two algorithms (Theys et al., 2021). It is expected that COBRA will replace the present operational $SO_2$ algorithm in near future. In this study, Level-2 COBRA data for the period from April 1, 2018, to December 31, 2021, were used. As for OMI and OMPS, clear-sky-only data, defined as having a cloud radiance fraction of less than 0.3, were used for the catalogue. Measurements at SZAs larger than 70° as well as measurements taken over snow or ice were excluded. The standard deviations

of individual TROPOMI $SO_2$ VCD were between 1 DU in the tropics and 1.5 DU at high latitudes for the original operational algorithm (Fioletov et al., 2020) and 50% lower for the COBRA version (Theys et al., 2021), i.e., comparable to or even lower than the OMI noise despite a 16 times smaller TROPOMI pixel area. Note that there is a systematic difference between OMI/OMPS and TROPOMI caused by the difference in $SO_2$ cross-section temperature (203K for TROPOMI vs. 293K for OMI/OMPS). To account for it, the TROPOMI $SO_2$ VCD values were increased by 22% (Fioletov et al., 2020; Theys et al.,

2017).

**2.3 Other data sets**

The emission estimation algorithm requires information on the wind speed and direction that are obtained from ECMWF reanalysis data. The most recent ERA5 wind data (C3S, 2017) provided U- and V- (west-east and south-north, respectively) wind-speed components with hourly temporal resolution on a 0.25° by 0.25° grid. They were grouped into 1 km-thick layers

and the mean wind speed and direction were calculated for each level as it was done for the original catalogue. The reanalysis wind data for regular pressure levels were linearly interpolated to overpass time and to the location of the centre of each satellite sensor pixel. The winds for the layer that corresponds to the height of a source were used by the algorithm. Please note, that in ERA5 for elevated locations, the wind data at levels below the surface pressure layer are simply duplicates of the winds at the lowest pressure available.

As in the original catalogue, measurements with snow on the ground were excluded from the analysis. The Interactive Multisensor Snow and Ice (IMS) Mapping System data were used as a source of the snow cover information (Helfrich et al., 2007).



## 3. Emissions estimates

### 3.1 The fitting algorithm

As mentioned in the Introduction, this study employs the same fitting algorithm as the original catalogue to estimate the total average $SO_2$ mass near the source and derive emissions assuming a constant lifetime. The algorithm is described in detail in our previous studies (Fioletov et al., 2015, 2016) and we briefly review its key features here.

The algorithm fits the plume from an emission point source by a fitting function and then this fit is used to calculate the total $SO_2$ mass ($\alpha$) near the source and the emission strength $E = \alpha/\tau$, where $\tau$ is a constant parameter that represents the

$SO_2$ lifetime (or decay time). The fitting function is an exponentially modified Gaussian (EMG) function (Beirle et al., 2014; Fioletov et al., 2015; de Foy et al., 2015) along the wind direction and a Gaussian function with plume width $\omega$ across the wind direction. The wind direction and speed are taken from the ERA5 reanalysis data. The algorithm uses two prescribed constant parameters ($\tau$ and $\omega$). One unknown parameter $\alpha$ is estimated from the fit. All satellite pixels within a rectangular area along the wind direction collected for one year (if annual emissions are estimated) are used for the fitting. The rectangular fitting

area extends $\pm L$ km across the wind direction, $L$ km in the upwind direction and $3 \cdot L$ km in the downwind direction, where the parameter $L$ depends on the emission strength on the source: from 30 km for sources under 100 kt yr$^{-1}$ to 90 km for sources greater than 1000 kt yr$^{-1}$.

The values of the prescribed parameters $\tau$=6 h and $\omega$=20 km, 25 km, and 15 km for OMI, OMPS and TROPOMI, respectively, are chosen as in the previous studies (Fioletov et al., 2020, 2016). Note, that the estimated emissions are not very

sensitive to small uncertainties in the plume width: e.g., a 5-km change (from 20 km to 25km) in the $\omega$ value for OMI produces only a 10%-15% change in the estimated emissions. As in previous studies, only pixels with associated wind speeds between 0.5 and 45 km h$^{-1}$ are used. Also, as in the original catalogue, high transient volcanic $SO_2$ VCDs were screened out by setting an upper limit on $SO_2$ VCDs: days when some pixels in the fitting area exceeded such limits were excluded from the analysis. These limits depend on the source emission strength (obtained from preliminary estimates).

There is a potential problem of overestimating emissions in the case of multiple sources in an area. Previous analysis of OMI data demonstrated that sources can be distinguished if the distance between them is greater than about 80 km but emissions can be overestimated if this distance is less than 50 km, although these limits would also depend on the emission strength and prevailing wind direction (Fioletov et al., 2016). In some cases, however, we fund that emissions from two sources can be clearly separated even if they are only 40 km apart. These limits should be even lower for TROPOMI data with its

much smaller pixel size. For this reason, we included 20 sources in the catalogue that are only 35-40 km apart but appear as isolated "hotspots" on TROPOMI $SO_2$ maps. For each source, we also added information on the distance to the nearest other source in the catalogue, so the users can do their own screening. In addition, to avoid "double counting" for regional emission estimates, some sources were not included in the catalogue if there is a source nearby that is already in the catalogue. This typically occurs in some regions of the US and China.





Based on the uncertainty budget of estimated emissions (Fioletov et al., 2016, their Table 1), the overall uncertainty is about 50%. The main contributors are uncertainties in AMFs, lifetime and plume widths that affect emissions as scaling factors, i.e., affect the absolute values but not so much relative changes of the estimated emissions. The $SO_2$ emission detection limit (defined as a level where the estimated annual emission is 3 times larger that its standard error) is about 8 kt $yr^{-1}$ for TROPOMI and 30–40 kt $yr^{-1}$ for OMI for the first years of operation and even higher for OMPS (Theys et al., 2021).

### 3.2. OMI/OMPS version 2

Version 2 OMI (and OMPS) VCDs are different from the previous version 1.2 that was used in the original catalogue. The differences are discussed first before we analyse the estimated emissions. The original OMI retrieval algorithm estimated $SO_2$ slant column density (SCD) first. The SCD was then converted to VCD by applying a constant AMF= 0.36 that was optimized for anthropogenic pollution in the eastern US in summer (Krotkov et al., 2008, 2006; Li et al., 2013). In the original $SO_2$

catalogue, we replaced that with source specific AMFs, which were pre-calculated using site-specific elevation, climatological aerosols, and surface reflectance (albedo) (McLinden et al., 2014). The same AMFs from the original catalogue were also applied to TROPOMI COBRA SCDs to calculate consistent VCDs and emissions (Theys et al., 2021). Although version 2 of OMI and OMPS data provide VCDs for each pixel (i.e., ColumnAmountSO2), we do not use them here to enshure consistency between the original and the new versions of the catalogue. The same is also true for TROPOMI COBRA $SO_2$ data.

To illustrate the differences between $SO_2$ VCDs calculated using different AMFs, **Figure 1** shows examples of $SO_2$ emissions from the original catalogue, emissions estimated from OMI Version 2 SCDs converted to VCDs using the same site-specific AMFs as in the original catalogue, and emissions estimated using Version 2 OMI VCD data (i.e., ColumnAmountSO2). We also included TROPOMI data that are discussed in section 3.4. **Figure 1** (**a** and **b**) shows estimated emissions from two Mexican sites, Tula and Cantarell, located at the same latitude and 7 degrees apart in longitude. Emissions

from Tula are well known (de Foy et al., 2009) and included in the CTM used to calculate a priori $SO_2$ profiles in the OMI Version 2 ColumnAmountSO2 product. In contrast, Cantarell is an oil field in the Gulf of Mexico (Fioletov et al., 2013; Villasenor et al., 2003) and its $SO_2$ emissions (mostly from flaring) may not be properly accounted for in the CTM. As a result, the a-priori $SO_2$ vertical profile, used in the Jacobian calculations assumes that the $SO_2$ is in the free troposphere, rather than near the surface and therefore overestimates the OMI sensitivity and leads to underestimation of emissions. Another example

of the difference between emission estimates from different data sets that is related to the source altitude is shown in **Figure 1** panels **c** and **d**. The two sources from the Middle East are at the same latitude, 5 degrees apart in the longitude, but at different elevations.

At present, the causes of all the differences between the estimates based on OMI/OMPS version 2 VCDs (i.e., ColumnAmountSO2) and SCDs (i.e., SlantColumnAmountSO2) using the source-specific AMFs is not always clear. In is

some cases, it was clear that the difference is related to missing emission information in the underlying CTM (as in the case of Cantarell), although in the others we do not have a good explanation. There are differences in the partial cloud correction is calculated between the two $SO_2$ algorithm versions; but analysis of emissions calculated for different cloud fraction thresholds





demonstrated that cloud fraction filtering cannot explain the difference. Another factor that could contribute to the difference is the two different versions of the reanalysis wind data sets, but the difference in the wind speed between the two reanalysis versions is on average within 1%-2% and is not enough to explain the difference in emissions.

As one of the goals of the catalogue is to improve the existing emissions inventories used in air quality models, we did not use version 2 VCDs that depend on the outputs of such models. Instead, the same approach as in the original catalogue was used, and source-specific AMF values were calculated.

It was found that even if we use the same AMF, emission estimated from OMI Version 2 SCDs are on average lower that the values from the original catalogue. To match the original catalogue values, we applied an empirical +10% correction to OMI and OMPS Version 2 SCD. As a result of this correction, the mean difference in annual emissions between the original catalogue and OMI emission estimates for the version 2 catalogues is less than 1%.

### 3.3. New site-specific AMFs

For version 2 of the catalogue, site-specific time-independent AMF values were used to calculate the $SO_2$ VCDs and emissions. A single AMF value is calculated for each source location using a similar approach as in the original catalogue (Fioletov et al., 2016; McLinden et al., 2016). AMFs were first calculated for a subsample of OMI observations (every 100[th] observation from every 3[rd] year, within 100 km of the source co-ordinates). Sampling in this way yields several thousand observations and is sufficient to represent conditions (cloud fraction, viewing and solar geometry, and seasonal sampling) for a given source for all three satellite instruments. The general approach from McLinden et al. (2014) is used, with one main exception. Here, the $SO_2$ profile is estimated based on the elevation of the source and a climatological boundary-layer height (as a function of latitude, longitude, month, and UTC hour) from von Engeln and Teixeira, (2013). Between these two altitudes the profile is assumed to be constant in mixing ratio, and zero elsewhere. The single, site-specific AMF is the average over these individual AMFs. In general, there is a relatively small difference between the original and new AMFs. The scatter plot of the AMFs used in the original catalogue vs. the AMFs used in the version 2 catalogue is shown in **Figure 2a**. The new AMFs are on average 10% higher (i.e., emission estimates are lower) for volcanic sources, about 5% higher for power plants and smelters, and practically unchanged for the oil and gas-related sources. The ratio of the new to the original AMF values is shown in **Figure 2b** as a histogram. On average, the ratio is slightly positive (1.07), it is between 1 and 1.13 for 50% of all sources and between 0.87 and 1.23 for 90% of sources. Differences between the old and new version are mainly a result of changes in the OMI effective cloud fraction.

### 3.4 TROPOMI COBRA

The TROPOMI COBRA $SO_2$ algorithm retrieves SCDs (Theys et al., 2021). To ensure consistency, we used the same approach as in the original catalogue: source-specific constant AMFs, although the AMF values were slightly different from the ones used in the original catalogue as discussed in Section 3.3. We also increase the TROPOMI VCDs by 22% as mentioned in Section 2.2.



### 3.5. The original vs. new emissions estimates

In summary, there are two main differences between the original and version 2 catalogue OMI-based emission estimates: (1) version 2 is based on a newer version of OMI $SO_2$ data and (2) it uses slightly different AMF values. To illustrate the differences between the two versions, **Figure 3** shows emission estimates from the original catalogue and those from the emission estimates of this study from three satellite instruments grouped by geographical region and source type. The comparison was done for the same 588 sources included in the most recent version of the original catalogue. The original catalogue and the new OMI-based emission estimates show, in general, a good agreement although the new OMI-based emissions are slightly higher for volcanic sources and lower over India. We observe no substantial regional patterns in the difference between the old and new OMI-based emission estimates. As for annual emissions from individual sources, the difference is within ±10% in 50% of cases and within ±23% in 90% of cases (only sources emitting >50 kt y$^{-1}$, were considered). **Figure 3** also shows emission estimates for the version 2 catalogue based on OMPS and TROPOMI COBRA data that are very similar to OMI-based estimates.

### 4.     The Version 2 catalogue

### 4.1 Merging the emission estimates

The emissions estimates were obtained using data from the three satellite instruments (OMI, OMPS and TROPOMI). In this section, we discuss some examples of these emission estimates for individual sources. In general, estimates from the different satellite instruments correctly capture long-term changes of emissions. As an illustration, **Figures 4a** and **b** shows the time series of annual emissions from two smelters. The Tsumeb smelter (19.23°S, 17.73°E), Namibia, was discussed by Ialongo et al. (2018). A sulfur-capture plant was installed there in 2015 to reduce $SO_2$ emissions and we see that estimates from all three satellite instruments show a two-fold decline in emissions thereafter (**Figure 4a**), although copper production has increased since 2015 (Ialongo et al., 2018). The overall emissions are relatively low, less than 100 kt y$^{-1}$, and OMPS data do not produce a reliable fit. This may explain why OMPS-based emissions are biased low. Another example is the Balqash smelter (46.83°N, 74.94°E), Kazakhstan, where $SO_2$ emissions also decreased from about 500 kt y$^{-1}$ in 2005 (that matches the available information on reported emissions from https://www.thegef.org/sites/default/files/ncsa-documents/2147-22347.pdf accessed on May 5 2022) to less than 100 kt y$^{-1}$ in 2010. A sulfur-capture plant became operational there on June 8, 2008 (http://www.kazakhmys.kz/ru/history accessed on May 5 2022) and substantially reduced the $SO_2$ emissions in the following years, but then all three satellite data sources show some increase in emissions after 2014.

One of the reasons for the difference between OMI, OMPS, and TROPOMI emissions estimates is the size of the source. The three satellite instruments have very different pixel sizes and, therefore, the source plumes observed by them have different widths. For isolated point sources, this should not affect the emission estimates since we use a constant instrument-specific plume width parameter as described in section 3.1. This is illustrated in **Figure 4c**, where emissions estimates from a



large source (Kluchevskoi volcano (56.06°N, 160.64°E), Russia) are shown. The source was highly variable in time, but all three instruments reported similar results. However, for sources that consist of multiple point sources with some distance between them, the aggregate plumes can be wider than the plume width used in the fit. **Figure 4d** shows emissions estimates from a cluster of sources in South Africa with nine power plants are located in an area of over 30 km in diameter. The assigned

plume width for TROPOMI and OMI (15 km and 20 km respectively) may not be large enough to describe the plume from that cluster of sources. As a result, estimated emissions from TROPOMI and OMI are lower than those from OMPS by 20% and 10%, respectively.  For non-point sources, the appropriate width parameter is influenced more by the spatial extent of the source and become independent of pixel size.  Thus, in the case of the South African cluster the width parameter would be the same for all three instruments, which would bring them into better relative agreement.

**Figures 4e** and **f** show an example of two sources 66 km apart. Erbil gas power station (36.01°N, 43.92°E) was developed in several stages: the first stage was completed in 2008, the next in 2011-2012, and then in 2014 (https://www.powermag.com/repowering-erbil-power-project-adds-500-mw-to-kurdistan-grid/ accessed on May 5 2022). The $SO_2$ emissions increased from about 20 kt y$^{-1}$ in 2005 to about 200 kt y$^{-1}$ in 2021. It is possible that this increase in emissions resulted in plumes from Erbil impacting Kirkuk (35.53°N, 44.34°E), located south-east of Erbil, leading to an increase of

estimated emissions for both Kirkuk and Erbil from low-resolution OMPS data in 2019-2021.

OMI and TROPOMI are the main instruments contributing to our emission estimates. The average value of the difference in annual emissions between the TROPOMI COBRA-based estimates and the OMI-based emissions estimates is less than 2%, the difference is within ±13% for 50% of cases and within ±28% for 90% of cases (for sources emitting >50 kt y$^{-1}$). This is an impressive result given that the instrument characteristics and retrieval algorithms are very different. It should

be reminded that the 10% correction of OMI and OMPS data and 22% correction of TROPOMI data discussed in section 3 were applied.

While emission estimates from three satellite instruments are available, it is more convenient for users to have a single emissions data set. It is also important for future continuation of the catalogue after the end of the Aura/OMI mission (expected before 2025). As such, the final version 2 catalogue emission values were calculated as weighted averages of emission

estimates from the three satellite instruments using inverse-variance weighting method; emission estimates for each satellite are weighted in inverse proportion to their variance (i.e., squared fitting uncertainty). The inverse-variance weighted average has the least variance among all weighted averages. The red line in the **Figure 4** shows such weighted averages, i.e., the version 2 catalogue values. For small sources, the variance of TROPOMI-based emission estimates is much lower than that that for OMI and OMPS. However, this difference diminishes for larger sources (Fioletov et al., 2020, their Figure 10). Therefore, the

contribution to the weighted average from different satellites depends on the source emission strength. While the actual weights are different from source to source and from year to year, the average weights are given in **Table 1**. As expected for small sources, the dominant contribution to the weighted average is from TROPOMI (about 90% for the sources below 30 kt y$^{-1}$), while for sources greater 300 kt y$^{-1}$, TROPOMI contributes less than 50%, with 33% contribution from OMI and 20% from OMPS.



Annual SO$_2$ emissions from the three satellite instruments and their weighted averages grouped by region and source type are shown in **Figure 5.** In general, all three satellite instruments agree well. In 2018-2021, the mean difference between OMI and TROPOMI estimates for all regions shown in **Figure 5** is within 10% except for the Former USSR (where it is about 12%) and the USA. In the case of the USA, the difference is about 40%. Most of the 57 USA sources were included in the catalogue due to their high emissions in the 2000s. Emissions declined in subsequent years due to scrubber installation and

most of the sources did not produce statistically significant emission estimates in 2018-2021. The OMI-based emission estimates for these sources in 2018-2021 were mostly random noise and, since we reported zero in case of negative emission values in the catalogue, their sum is biased high. If we only consider sources that emitted more than 30 kt y$^{-1}$ in 2018-2021, then the difference between TROPOMI and OMI-based estimates is below 10%. OMPS-based emissions estimates are also slightly higher in some regions for the same reason. As the weights for OMPS are the lowest, their contribution to the weighted

average is small. Finally, the difference between the weighted average (i.e., the values from the version 2 catalogue) and mean TROPOMI-based emissions averaged for all 8 regions shown in **Figure 5** is between -4% and +3%, and for OMI and OMPS it is within ±12% (for sources > 30 kt y$^{-1}$).

**4.2 New sources and new types of sources**

In this section we discuss some changes in the sources listed in the catalogue and their emissions. The original catalogue

contained emission estimates for 491 sources. More sources have been added since then and the 2019 update of the catalogue contained 555 sources. The Version 2 catalogue contains 759 sources: 477 power plants, 74 smelters, 102 oil and gas-related sources and 106 volcanos. A map of the sources and the catalogue evolution is shown in **Figure 5**. The main reason for adding more sources was a lower emission uncertainty when TROPOMI data are used that made it possible to monitor smaller sources. In addition, better databases of industrial source locations made it possible to identify more sources in past data. Four sources

(Severodvinsk, Serov, Turov, and Fushina) were excluded from the original version because their emissions fell below the significance level when version 2 OMI data were used. Some of the sources from the original catalogue did not produce significant emissions during the TROPOMI period (2018-2021): the maximum ratio of estimated annual emissions to their standard deviation was less than 3 for 62 sources and less than 5 for 125 sources.

  There have been numerous changes in emissions from the listed sources since publication of the original catalogue.

There is an overall decline in emissions from the US, Europe and China as illustrated by **Figure 6**. This is largely due to installation of sulfur-capturing devices at power plants in these regions. There is also an overall decline in emissions from smelters. Some smelters, e.g., the Flin-Flon and Thompson smelters (Canada), have been closed, while the others, e.g., Bor (Serbia), Tsumeb (Namibia), have installed scrubbers that reduced SO$_2$ emissions (Ialongo et al., 2018). While there is a decline in emissions from the oil and gas-related sources listed in the original catalogue (**Figure 3**), there are also a number of new

such sources. As a result, overall emissions from this source type remain almost unchanged (**Figure 6**).

  There are 13 and 43 additional sources in India and China, respectively, mostly coal-burning power plants. Many of them were built earlier, but not included in the original catalogue because it was difficult to properly identify them. The recently



released power plant databases made this identification easier. These additional sources increased the estimated total emissions from China and India.

There are 38 additional sources (power plants and oil and gas processing facilitied) in the Middle East region. The Al-Khairat power plant (32.43°N, 44.28°E), Iraq, is one of such examples. It was built in 2013 and both OMI and TROPOMI-based estimates show a persistent emission of about 170 kt y$^{-1}$ in 2014-2021. Another example is a power and desalination plant in Shuqaiq (17.66°N, 42.08°E), Saudi Arabia (http://sqwec.com/), developed in multiple phases. Operations started in 2010 and emissions have increased rapidly since 2016 reaching 300 kt y$^{-1}$ in 2021.

In the catalogue, there are three categories of industrial $SO_2$ sources: power plants, smelters, and oil and gas sector-related sources. There are, however, some sources that do not fall under any of these categories. One such source is a cluster of small ceramic factories at Morbi (22.8°N, 70.9°E), India, that was discussed in detail by Kharol et al. (2020). Available emissions inventories do not report any major sources in this region, and yet this source with emissions of about 100 kt y$^{-1}$ is one of the largest in the area. Another example is a large cluster of brick kilns near Dhaka (23.63°N, 90.45°E), Bangladesh.

These sources are included in the version 2 catalogue. We decided to list them under the "power plant" category rather than create an additional category.

Cement production is also a source of $SO_2$ emission, where it is produced from coal combustion (Reddy and Venkataraman, 2002). Emissions from cement plants are too small, less than 50 kt y$^{-1}$ to be detected by OMI, but two such sources (Shree (26.3°N, 74.13°E), India and Thap Kwang (14.63°N, 101.08°E), Thailand) can be detected by TROPOMI and

are  included in Version 2 catalogue. Each of these sources is a cluster of several individual cement factories. We did not introduce a new source type in the catalogue and assigned the "power plant" source type for these sources (but included this information in the comment column).

It is rare that new large emissions sources appear at high latitudes. Two examples are production plants in the Russian Arctic: Bajandyskaya (66.432°N, 56.6°E) and East Lambeishor (66.764°N, 56.192°E) (https://energybase.ru/compressor-

station/oil-treatment-plant-opf-east-lambeishor accessed on Jan 26, 2022) that began their operation in 2014 and 2017, respectively. The plants are 42 km apart and they process the fluid mixture of oil, gas, and water from oil wells, remove hydrogen sulfide and prepare the oil for further use. We assume that the main $SO_2$ emission sources are the gas flares that are clearly visible on satellite images displayed in Google Maps. The TROPOMI-based emissions are about 110 kt y$^{-1}$ for Bajandyskaya and about 70 kt y$^{-1}$ for Lambeishor, although there could be some double-counting of emission due to their close

proximity. OMI data are too noisy for reliable emission estimates from these two sources.

Emissions sources can often be detected from multiyear mean $SO_2$ VCD maps and then confirmed using high resolution satellite imagery (Dammers et al., 2019; Fioletov et al., 2016; McLinden et al., 2016). Satellite images helped us to link some $SO_2$ hotspots with powerships, i.e., power plants installed on a moving platform, like a ship. One such source was identified at the port of Dakar (14.69°N, 17.43°W), Senegal, and is included in the new catalogue. Karpowership's powership

with 235 MW capacity was deployed there in October 2019 (https://karpowership.com/en/project-senegal accessed on Jan 26, 2022). The estimated emissions are about 40 kt y$^{-1}$ but it is likely the combined emissions from the powership and the existing




power plant in the area. Another example is Port de Mariel (23.02°N, 82.75°W), Cuba, where three powerships with a total capacity of 184 MW were installed in November 2019 (https://karpowership.com/en/project-cuba accessed on Jan 26, 2022) in addition to the existing power plant. As a result, total emission from Mariel increased from about 70 kt y$^{-1}$ to about  90 kt

y$^{-1}$. The contribution of such powerships to the total national electricity needs in both these cases are rather substantial, about 10%. Large powerships have also been in operation at Zouk (33.96°N, 35.61°E,) and Jieh (33.65°N, 35.4°E), Lebanon since 2013, each with a capacity of 202 MW (https://karpowership.com/en/lebanon accessed on Jan 26, 2022). The powership at Jieh is included in the new catalogue and its emissions are estimated to be about 20-30 kt y$^{-1}$. In the case of Zouk, the powership was located near a power plant that has already been included in the original catalogue.

Version 2 catalogue emission estimates for OMI and, particularly, TROPOMI COBRA show a lower noise level in the South Atlantic Anomaly area. This made it possible to estimate emissions from two volcanic sources (Planchón-Peteroa (35.27°S, 70.57°W), Argentina and Lascar (23.37°S, 67.68°W), Chile) and from four oil and gas-related sources in Brazil.

Six new volcanic sources were added to the catalogue. In addition to the two mentioned above, there are three others located in Alaska (Makushin), the Kuril Islands (Ebeko), and Nicaragua (Momotombo). Their maximum annual emissions

range from 50 to 150 kt y$^{-1}$. We also added La Palma volcano, Canary Islands (https://volcano.si.edu/volcano.cfm?vn=383010 accessed on May 26, 2022) that became active in 2021, although most of SO$_2$ detected there was emitted during a major eruption in September-December 2021 and therefore not reflected in the catalogue due to the screening of high transient volcanic SO$_2$ VCDs.

### 4.3 Emissions by region and source type

**Figure 7** illustrates the emission sources listed in the catalogue at the beginning (2005) and the end (2021) of the available data period. The symbol size is proportional to the emission strength, while the colour represents the source type. Industrial sources with annual emissions under 3 standard errors (σ) of the estimate are not shown. There is a clear decline in the number of detectable sources over the U.S., China, and Europe despite three times lower emission estimates uncertainties for small sources due to TROPOMI data in recent years. For example, only 11 industrial sources produced emissions above 3σ level in

the US in 2021, while there were 57 such sources in 2005. There are several clusters of sources visible on the **Figure 7** maps: power plants in India and China, oil and gas-related sources in the Middle East and a number of smelters along the west coast of South America.  The largest sources such as Norilsk and the cluster of power plants in South Africa demonstrated little changes in their emissions.  **Figure 7** also shows an increase in the number of sources in India and the Middle East.

While the absolute values of emissions are shown in **Figure 7**, the relative contribution to total emissions in different

years grouped by the source type and country/region is shown in **Figure 8**. The decline in emissions from China, the U.S., and Europe was largely related to the power generation sector. As a result, total SO$_2$ emissions from power plants declined by about 60% and their relative contribution to total SO$_2$ emissions has declined since 2005 from 52% to 44%.  Emissions from smelters demonstrated a similar decline and the relative contribution of smelters remains unchanged (10%-12%). However,



emissions from oil and gas-related sources do not show any changes and the fraction of emissions related to the oil and gas

industry has increased from 11 to 17%.

On a global scale, the new catalogue data show an approximate 50% decline in global $SO_2$ emissions between 2005 and 2021, although the $SO_2$ emissions appear to have levelled off before 2008, between 2009 and 2013 and in the last 3 years. On a regional level, there has been a remarkable decline in emissions from the U.S. and Europe since 2005. Their emissions also levelled off in the last 3-4 years. The relative contributions of the U.S. and European emissions to the total anthropogenic

emissions are about 2% and 3.5% respectively. Emissions from China increased at the beginning of the record, but they declined thereafter. China contributed nearly 40% to total anthropogenic emissions in 2005-2010, and its contribution declined to under 11% in 2020. Emissions from India have surpassed emissions from China after 2015 (Li et al., 2017), but they levelled off in recent years. At present they account for 15% of the total anthropogenic emissions. Emissions from the Middle East show some increase, although they also levelled off in the recent years. Nevertheless, their relative contribution increased from

13% in 2005 to 24% in 2020. The former USSR countries that include Russia, Ukraine, Kazakhstan, and former USSR countries in Central Asia, also demonstrated some decline in emissions, but some increase (from 11% to 17%) in their relative contribution.

Degassing volcanoes contribute between one quarter and one third of total $SO_2$ emissions. After remaining roughly constant for the first 15 years or so, total volcanic emissions started declining in 2018 and were lower by up to 40% in 2019-

2021. This was mainly due to a very large decline (1000-2000 kt y$^{-1}$) in emissions from some strong volcanic sources including Kilauea (Hawaii) and Aoba and Ambrym (Vanuatu) volcanoes.

## 5.     Summary and discussion

To update the original catalogue of large $SO_2$ sources and emissions (Fioletov et al., 2016), a new version 2 of the catalogue was developed that merges the data from three satellite instruments: OMI, OMPS, and TROPOMI for the 2005-2021 period.

For OMI and OMPS, version 2 $SO_2$ data (Li et al., 2020c) were used. For TROPOMI, the COBRA research data product (Theys et al., 2021) was used. For all data sets, SCDs were converted to VCDs using a set of site-specific AMFs as in the original catalogue, although for catalogue version 2 the AMFs were recalculated using the most recent data on albedo and climatological aerosol data. As in the original catalogue, only measurements under snow/ice-free and mostly cloud-free conditions were used in the analysis. The total number of sources in the Version 2 catalogue is 759, including 477 power

plants, 74 smelters, 102 oil and gas-related sources, and 106 volcanos. However, some of these sources were not active in recent years: 62 sources from the original catalogue do not produce detectable emissions in 2018-2021. Four sources were excluded from the original catalogue because their, estimated emissions using version 2 OMI data are below the significance level.

The $SO_2$ emission estimation algorithm is identical to that used in the original study (Fioletov et al., 2016) to assure

the continuity of the old and new emissions estimates. Unlike the original catalogue, where ERA-Interim reanalysis wind data



were used, the new catalogue employed ERA5 reanalysis data. For consistency with the original catalogue, the values obtained from version 2 OMI and OMPS-based estimates were increased by +10%. For TROPOMI, a +22% correction was applied to account for differences in temperatures for the $SO_2$ absorption coefficients used in the retrievals. Note that the original catalogue also had empirical corrections applied to ensure agreement with reliable at stack emission measurements in the U.S.

and other countries (Fioletov et al., 2016).

The Version 2 catalogue emissions are weighed averages of OMI, OMPS, and TROPOMI-based emission estimates using an inverse-variance weighting method. If emission estimates from all three satellite instruments are available, the TROPOMI based estimates dominate in the weighted average for small sources (about 90% contribution for sources under 30 kt y$^{-1}$). For large sources (300 kt y$^{-1}$), contributions from TROPOMI, OMI, and OMPS-based emission estimates were 47%,

33%, and 20% respectively.

As previously discussed (Fioletov et al., 2016; Theys et al., 2021), the emission detection limit is typically about 30-40 kt y$^{-1}$ for OMI data and 8 kt y$^{-1}$ for TROPOMI data, although these values vary from source to source depending on the AMF values and local conditions.

Systematic differences between the emission estimates from OMI, OMPS, and TROPOMI data are less than 12% for

all eight large regions analysed in this study. However, the difference could be larger for individual sources. The differences in annual emissions between the TROPOMI COBRA-based estimates and the OMI-based estimates are within ±13% for 50% of cases and within ±28% for 90% of cases (only sources emitting >50 kt y$^{-1}$, were considered). Large differences are typically seen for sources comprised of several individual point sources. Sources in close proximity are one of the main obstacles to reliable emission estimations when using the point-source emission algorithm. Emissions from sources 60-80 km apart

typically can be reliably estimated, while sources under 20-30 km apart are counted as a single source. However, it also depends on the wind climatology and emissions strength of the sources. For user convenience, for each catalogue entry, we included information about the nearest other catalogue source and the distance to that source.

The original catalogue was successfully used to improve emission inventories used in air quality and climate models, and to monitor emission reductions due to sulfur-capturing device installation and other applications. The version 2 catalogue

updates the emission estimates using the most recent version of OMI $SO_2$ data and utilises emission estimates from two other operational UV satellite instruments, OMPS and TROPOMI. For user's convenience, the version 2 catalogue data set also contains emission estimates for individual satellite instruments that can be further used for analysis and cross-validation of the different satellite data sources.

## 6. Data availability

The full version 2 $SO_2$ point source catalogue is available from the NASA Global Sulfur Dioxide Monitoring Home Page (https://so2.gsfc.nasa.gov/measures.html accessed on August 11, 2022). The direct link to the data set is



https://so2.gsfc.nasa.gov/kml/Catalogue_SO2_2022.xls.                    DOI            identifier            is
https://disc.gsfc.nasa.gov/datasets/MSAQSO2L4_1/summary (DOI:

10.5067/MEASURES/SO2/DATA403 – the present version 1 will be replaced with version 2 if the paper is accepted)

The TROPOMI COBRA SO$_2$ data set is available from co-author Nicolas Theys on request. The OMI and OMPS PCA SO$_2$ data are publicly available from the Goddard Earth Sciences (GES) Data and Information Services Center (DISC) (https://disc.gsfc.nasa.gov/datasets/OMSO2_003/summary accessed on August 8, 2022 (Li et al., 2020a) and https://disc.gsfc.nasa.gov/datasets/OMPS_NPP_NMSO2_PCA_L2_2/summary accessed on August 8, 2022 (Li et al., 2020b)).

**Author Contributions.** VF prepared the paper and figures with contributions from all the co-authors. VF and CM developed the emission estimation algorithm, VF, CM and DG processed satellite data and estimated the emissions. CL, NK, and JJ developed the OMI and OMPS PCA version 2 algorithms and provided data. NT developed the TROPOMI COBRA algorithms and provided data. IA developed some of the emission data visualization tools. PL provided support for hosting the catalogue. SC contributed to cataloguing of volcanic sources and interpreted the estimated volcanic emissions.

**Competing interests.** The authors declare that they have no conflict of interest.

**Acknowledgments**

We thank Keith Evans for developing SO2 emissions web site: https://so2.gsfc.nasa.gov/measures.html. We also thank EU/ESA/KNMI/DLR for providing the TROPOMI/S5P Level 1 products. Can L, Nickolay Krotkov, and Joanna Joiner acknowledge support from the NASA Earth Science Division Aura Science Team, Suomi NPP Science Team and US
participating investigator programs. OMI PCA SO$_2$ data used in this study have been publicly released as part of the Aura OMI Sulfur Dioxide Data Product - OMSO2 and can be obtained free of charge from the Goddard Earth Sciences (GES) Data and Information Services Center (DISC, http://daac.gsfc.nasa.gov/). Nicolas Theys acknowledges support from ESA (S5P-PAL and S5P MPC projects) and BELSO Prodex-Trace S5P.





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





**Table 1**. Relative contribution of individual satellite instruments to the weighted average for emissions estimate depending on the emission strength for 2018-2021.

| Source emissions (kt/year) | | Relative contribution (%) | | |
|---|---|---|---|---|
| From | To | OMI | OMPS | TROPOMI |
| 0 | 30 | 7 | 5 | 88 |
| 30 | 100 | 13 | 8 | 79 |
| 100 | 300 | 25 | 15 | 59 |
| 300 | 1000 | 33 | 20 | 47 |
| 1000 | 3000 | 34 | 20 | 45 |
**Figure 1.** Four examples of SO$_2$ emission estimates with 2$\sigma$ error bars from the original catalogue (red), emissions estimated from OMI Version 2 SCD values and converted to VCDs using the same AMFs as in the original catalogue (blue), and emissions estimated using Version 2 OMI VCD data (green). We also included TROPOMI-based estimates (cyan). The emission source names, coordinates, and elevations above sea level are shown. TROPOMI-based estimates were increased by 22% and OMI and OMPS data were increased by 10% as discussed in the text.

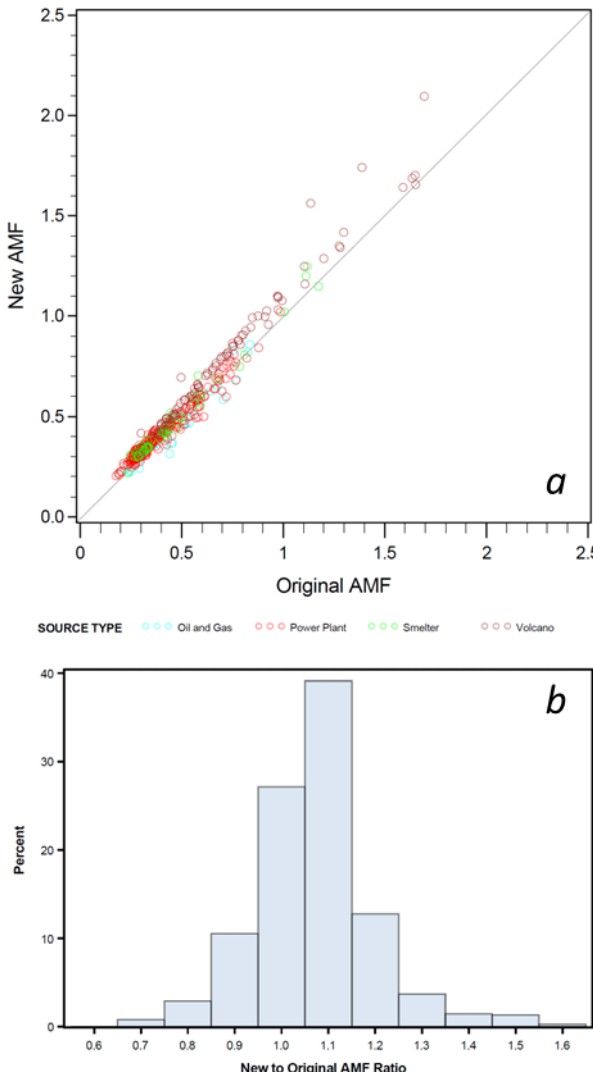

**Figure 2.** (a) Scatter plot of the AMFs used in the original catalogue vs. the AMFs used in the version 2 catalogue. Each dot corresponds to one emission source and the dot color reflects the source type as shown in the legend. There are total of 555 sources on the plot. The correlation coefficient between the two data sets is 0.98. (b) The distribution of the ratio of the new AMF values to the AMF values in the original catalogue.




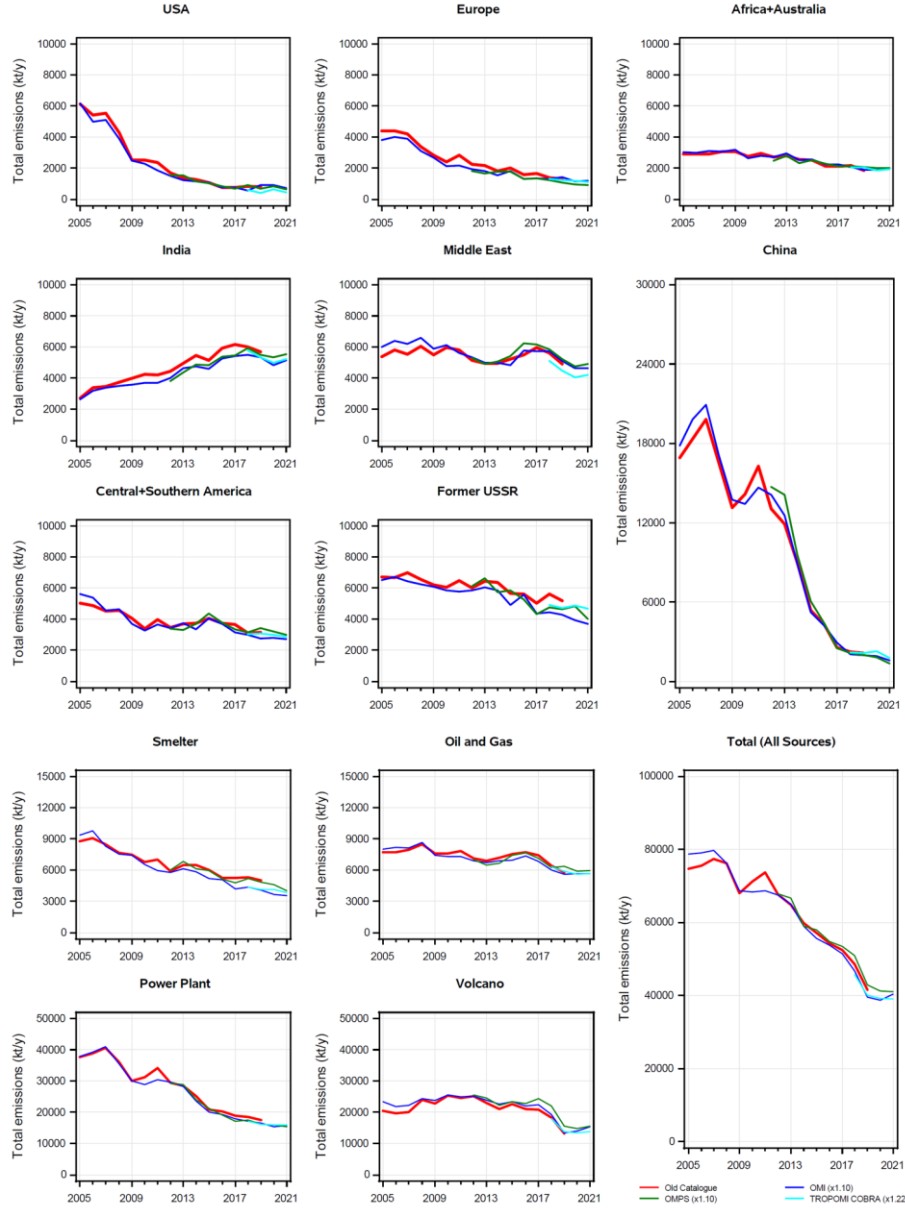

**Figure 3.** Annual SO$_2$ emissions estimated from four satellite data sets: the original catalogue (red), OMI-based estimates for
the version 2 catalogue (blue), OMPS-based estimates for the version 2 catalogue (green), and TROPOMI COBRA-based
estimates for the version 2 catalogue (cyan). The data are grouped by region (8 top panels), by emission source type (bottom
4 panels) and the bottom right panel shows total emissions from all sources. TROPOMI-based estimates are adjusted upwards
by 22% to account for the difference in used SO$_2$ cross-sections and OMI and OMPS data are adjusted upwards by 10% as
discussed in the text. Only sources included in the original catalogue with the original AMF values were used in this
comparison.

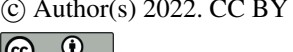



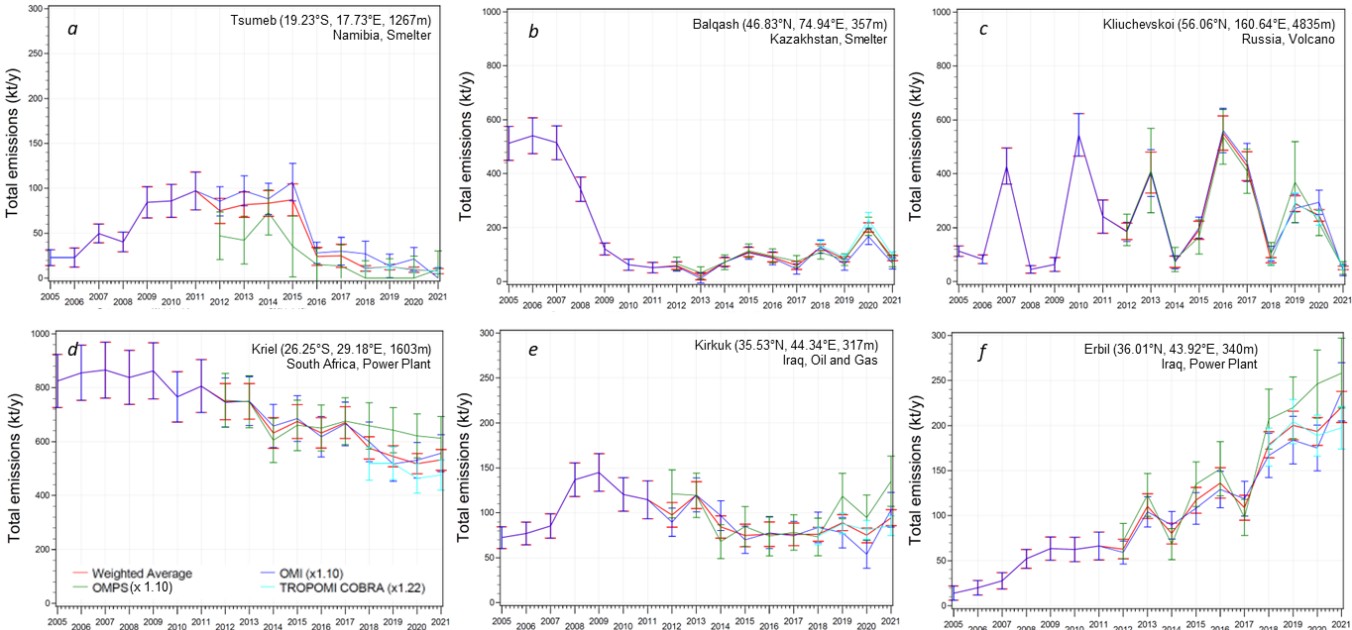

**Figure 4** Annual emission estimates from the three satellite data sources: OMI (blue), OMPS (green) and TROPMI (cyan), and their weighted average (red) with 2σ error bars. TROPOMI-based estimates were increased by 22% and OMI and OMPS
data were increased by 10% as discussed in the text. The emission source names, types, coordinates, and elevations above sea level are shown.



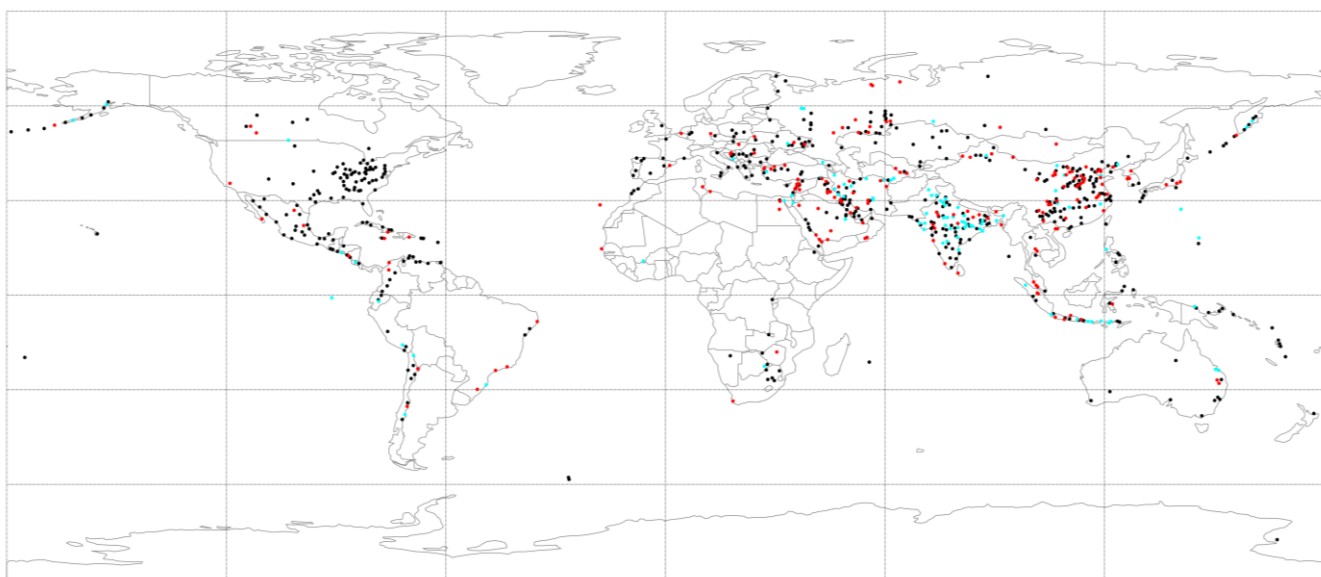

**Figure 5.** Map of the sources included in the version 2 catalogue. The sources included in the original publication (Fioletov et al., 2016) are shown as black dots, the sources added to the catalogue in 2017-2019 are shown as blue dots, and sources added in the version 2 catalogue are shown as red dots.



**Figure 6.** Annual SO$_2$ emissions from four satellite data sets: version 2 catalogue OMI VCD data (blue), version 2 catalogue OMPS data (green), version 2 catalogue TROPOMI COBRA data (cyan) and the weighted average (red). The data are grouped by region (8 top panels), by emission source type (bottom 4 panels) and the bottom right panel shows total emissions from all sources. TROPOMI-based estimates were increased by 22% and OMI and OMPS data were increased by 10% as discussed in the text. The weighted average is based on OMI data only in 2005-2011, on OMI and OMPS data in 2012-2017, and on data from all three instruments in 2018-2021.



735

**Figure 7.** SO₂ emissions sources listed in the version 2 catalogue for 2005 (top) and 2021 (bottom). Only sources where the ratio of the emission value to its standard deviation is greater than 3 are shown. The size of the symbols is proportional to the annual emission values.





**Grouped by source type**

**2005** 52.37% 10.66% 25.18% 11.79%

**2010** 47.65% 11.31% 31.15% 9.89%

**2015** 40.91% 13.97% 35.09% 10.03%

**2020** 44.13% 16.78% 28.05% 11.03%

☐ Oil and Gas   ■ Power Plant   ☐ Smelter   ☐ Volcano

**Grouped by region**

**2005** 37.98% 12.69% 10.78% 8.95% 8.30% 6.89% 5.5% 4.82% 4.10%

**2010** 37.58% 15.63% 12.34% 8.77% 7.00% 5.84% 5.10% 4.03% 3.72%

**2015** 19.63% 21.45% 15.37% 11.31% 9.76% 9.53% 6.30% 4.23% 2.42%

**2020** 24.17% 17.21% 15.36% 12.13% 10.84% 8.70% 6.16% 2.56% 1.86%

☐ Africa+Australia   ■ Central+Southern America   ■ China
☐ Europe   ☐ Former USSR   ☐ India
☐ Middle East   ☐ USA   ☐ Other

740

**Figure 8**. (Top) Pie charts of contributions to total $SO_2$ emissions by the source type. Power plants are the main source of emissions and degassing volcanoes contribute between one quarter to one third of total emissions. (Bottom) Pie charts of contributions to total anthropogenic $SO_2$ emissions by region.