# Peer review of "Version 2 of the global catalogue of large anthropogenic and volcanic SO2 sources and emissions derived from satellite measurements"

_Earth System Science Data, 2022_

## Referee Comment (RC1)

Review of the manuscript: " Version 2 of the global catalogue of large anthropogenic and volcanic SO2 sources and emissions derived from satellite measurements" by Fioletov et al.

General comments:

The manuscript presents an update of the SO2 emission catalogue based on SO2 satellite observations. The new dataset includes updates in the retrieval algorithm, more accurate wind information and synergistic use of different satellite observations. Such update is very welcome since this emission dataset is quite useful for both scientific and societal applications. The method is scientifically sound, and I recommend publication after addressing the following minor issues.

Specific comments:

- Concerning the merging of the different emissions, it was not completely clear how the different instruments contributions are applied when you have only one or two instruments/estimations available. I mean before TROPOMI for example, are the estimates mostly based on OMI? And do you see a bias when introducing TROPOMI estimates into the merged estimates compare to OMI+OMPS only? Please clarify.
- Connected to question n.1, what happens to sources you only detect with TROPOMI: do you have zero emission before the TROPOMI period, or do you attempt the fitting with OMI/OMPS anyways even if the detection limit is higher? For example, the two Russian arctic sites you mention have emission estimates in the database also before the TROPOMI period, even though you write that those are not reliable: can you clarify?
- Is there a chance to attribute some of the time series flattening in India to COVID-related issue?
- Is there a reason you put together former USSR countries? Do for example trends in eastern Europe or Central Asia look the same than Russia? I would expect different policies in terms of emission regulation in these different countries.

Technical comments

Line 114 "epy": what do you mean?

L183 "fund" -> "found" or "find"

---

## Referee Comment (RC2)

Review of Fioletov, et.al., Version 2 of the global catalog of large anthropogenic and volcanic SO2 sources and emissions derived from satellite measurements.

**General Comments**

This paper describes a significant expansion of the 2016 OMI-derived global catalog of $SO_2$ emission sources enabled by the launch of new UV mapping instruments; OMPS on SNPP, the first operational instrument of this type, and TROPOMI, similar to OMI, but with much higher ground resolution. In addition, new retrieval algorithms for all three instruments offer significant improvements, such as lower detection thresholds. As a result, twice as many sources were found for this Version 2 catalog.

The introduction summarizes the history of UV $SO_2$ measurements from LEO since Nimbus-7 SUV/TOMS in 1978. TOMS, limited by telemetry bandwidth to 6 wavelengths (selected for total ozone determination), and relatively crude spatial resolution, given 1970s satellite technology, could measure large volcanic eruption plumes but had limited ability to observe surface sources of $SO_2$. GOME and SCIAMACHY similarly focused on ozone with its low spatial variability, but, with far more spectral information could detect some surface $SO_2$ sources. However, OMI addressed an essential factor for the detection of point sources of $SO_2$ with an 8-fold higher ground resolution than TOMS. The 2016 $SO_2$ catalogue of OMI data, updated annually and publicly available on a NASA website, discriminated four sources of large emissions; powerplants, smelters, the oil and gas industry, and volcanoes. This first-ever assessment of industrial $SO_2$ emissions from space showed the advantage of satellite monitoring as a uniform, independent source of information. The reference list is extensive and appears to be complete.

The satellite record of global sulfur dioxide emissions and the changes over time since 2005 is an important contribution to the assessment of air pollution and passive volcanic emissions. Having multiple, well-characterized, redundant satellite data Is a real asset in assessing errors. Fioletov and colleagues deserve praise for their efforts to produce best estimates of emissions from these somewhat diverse data sources. Analyzing the differences is a large effort considering retrieval algorithm changes, retrieval uncertainties, air mass factors, differences between coincident observations, quality control data selection standards, among other factors. Merging the data required difficult choices. The $SO_2$ total emission from each source is then calculated by combining the overpass sample of the plume mass with ECMWF reanalysis wind information and loss rate estimates. All of these considerations are documented in the text and figures.

The introduction of two additional satellite data sources offers new challenges as described in the text. Inevitable differences require explanations that should lead to a better understanding of the measurement and retrieval methods.

**Specific Comments**

Because the retrieval algorithms derive slant column $SO_2$ amounts an air mass factor is required to convert them to a quantitative geophysical measure, namely the vertical column $SO_2$ amount. The differences between the AMF choices almost suggest that this is a dark art. The authors make clear that the height of the absorbing layer is a primary factor affecting the AMF, although other factors such as partial cloud corrections also play a role. They chose to use independently calculated site-specific AMFs, similar to the original catalogue methods. The procedure is documented and referenced.

The need for arbitrary corrections (Ie, 22%) is disturbing. It's surprising that the TROPOMI data production group uses $SO_2$ absorption cross sections suitable for tropopause level air temperatures when a primary mission objective is to measure surface and low altitude plumes. Theys, et al. (2017) contend that this is "in principle accounted for in the AMFs". So why do we need a 22% correction?

It is concerning that the OMI data are empirically corrected by 10% to force agreement with the original catalogue results. Please explain the rationale for this correction. Are more ground-truth estimations now available since the first catalogue? A discussion of the ground truth analysis would be appropriate especially since the original catalogue had empirical corrections to "ensure agreement with reliable at stack emissions measurements…".

The procedures for merging the three data sets are well documented. The difficulties of multiple-source regions and the effect of ground resolution differences are described.

Lumping the former USSR countries together is no longer meaningful.

The reference list is extensive.

**Technical corrections**

Line 183 sp      fund => find
Line 203  sp     enshure => ensure
Line 275  ?   "….and OMPS data do not produce a reliable fit. This may explain why OMPS-based emissions are biased low." This requires a bit of explanation.
Line 313.  Repeat        "that that"
L. 321, 322, 331:  "Figure 5" is incorrect.  Figure 6?

L 386 – 7.  "The estimated emissions are about 40 kt $y^{-1}$ but it is likely the combined emissions from the powership and the existing power plant in the area."  Incomplete sentence.

L 431.   increase => increased
L 456  typo  weighed = > weighted

L 475  typo   utilises => utilizes

L 498   "Can L" should be "Can Li"

I suggest running a spell check on the document.  I doubt that I have caught all the errors.

---

## Author Comment (AC1)

Review of the manuscript: " Version 2 of the global catalogue of large anthropogenic and volcanic SO2 sources and emissions derived from satellite measurements" by Fioletov et al.

General comments:

The manuscript presents an update of the SO2 emission catalogue based on SO2 satellite observations. The new dataset includes updates in the retrieval algorithm, more accurate wind information and synergistic use of different satellite observations. Such update is very welcome since this emission dataset is quite useful for both scientific and societal applications. The method is scientifically sound, and I recommend publication after addressing the following minor issues.

We would like to thank the reviewer for his favorable comment.

Specific comments:

• Concerning the merging of the different emissions, it was not completely clear how the different instruments contributions are applied when you have only one or two instruments/estimations available. I mean before TROPOMI for example, are the estimates mostly based on OMI? And do you see a bias when introducing TROPOMI estimates into the merged estimates compare to OMI+OMPS only? Please clarify.

We added a paragraph that discussed this issue:

"*Prior to 2012, only OMI data were available, and the weighted average was just OMI-based emissions. In 2012-2017, the weighted average of OMI and OMPS was used. Some sources in some years did not have enough data to produce estimates from OMPS and in such cases, the average was based on OMI data only. Although statistically significant annual emissions estimates for some sources can be obtained from TROPOMI data only, we nevertheless included OMI and OMPS-based estimates in the weighted average for such sources in the catalogue. Multiyear averages for such sources could be significant even prior to the TROPOMI measurements.*"

As for possible OMI/OMPS-TROPOMI biases, they are discussed in the manuscript: "*the difference is within ±13% for 50% of cases and within ±28% for 90% of cases*". The biases are even smaller when summing sources over large regions (Figure 6). There are some sources, where the difference between TROPOMI and OMI are noticeable, and their possible causes are discussed in the text. We prefer not to adjust emission estimates for individual sources to remove a possible TROPOMI-OMI bias. Instead, we made estimates for individual satellites available.

• Connected to question n.1, what happens to sources you only detect with TROPOMI: do you have zero emission before the TROPOMI period, or do you attempt the fitting with OMI/OMPS anyways even if the detection limit is higher? For example, the two Russian arctic sites you mention have emission estimates in the database also before the TROPOMI period, even though you write that those are not reliable: can you clarify?

We added a paragraph that discussed this issue (see the previous comment). Annual emissions can be calculated even if the emissions are below the detection limit. It just means that such estimates would be within their uncertainties. However, such estimates could be useful to calculate multiyear averages, that could be above the uncertainties.

• Is there a chance to attribute some of the time series flattening in India to COVID- related issue?

We see small decline in Indian emissions in 2020 (Figure 5). However, we estimated annual emissions, while the most COVID-related decline likely occurred over approximately one season.

• Is there a reason you put together former USSR countries? Do for example trends in eastern Europe or Central Asia look the same than Russia? I would expect different policies in terms of emission regulation in these different countries.

We change that to "northern Eurasia" (Russia, Ukraine, Kazakhstan, and former USSR countries in Central Asia). To our best knowledge, no regulation to reduce power plant emissions was introduced in these countries. This makes that region different from, for example, Estonia where emissions have been reduced substantially since Estonia joined the EU.

It was a typo. Removed.

Corrected to "found".

---

## Author Comment (AC2)

Review of Fioletov, et.al., Version 2 of the global catalog of large anthropogenic and volcanic SO2 sources and emissions derived from satellite measurements.

**General Comments**

This paper describes a significant expansion of the 2016 OMI-derived global catalog of SO2 emission sources enabled by the launch of new UV mapping instruments; OMPS on SNPP, the first operational instrument of this type, and TROPOMI, similar to OMI, but with much higher ground resolution. In addition, new retrieval algorithms for all three instruments offer significant improvements, such as lower detection thresholds. As a result, twice as many sources were found for this Version 2 catalog.

The introduction summarizes the history of UV SO2 measurements from LEO since Nimbus-7 SUV/TOMS in 1978. TOMS, limited by telemetry bandwidth to 6 wavelengths (selected for total ozone determination), and relatively crude spatial resolution, given 1970s satellite technology, could measure large volcanic eruption plumes but had limited ability to observe surface sources of SO2. GOME and SCIAMACHY similarly focused on ozone with its low spatial variability, but, with far more spectral information could detect some surface SO2 sources. However, OMI addressed an essential factor for the detection of point sources of SO2 with an 8-fold higher ground resolution than TOMS. The 2016 SO2 catalogue of OMI data, updated annually and publicly available on a NASA website, discriminated four sources of large emissions; powerplants, smelters, the oil and gas industry, and volcanoes. This first-ever assessment of industrial SO2 emissions from space showed the advantage of satellite monitoring as a uniform, independent source of information. The reference list is extensive and appears to be complete.

The satellite record of global sulfur dioxide emissions and the changes over time since 2005 is an important contribution to the assessment of air pollution and passive volcanic emissions. Having multiple, well-characterized, redundant satellite data Is a real asset in assessing errors. Fioletov and colleagues deserve praise for their efforts to produce best estimates of emissions from these somewhat diverse data sources. Analyzing the differences is a large effort considering retrieval algorithm changes, retrieval uncertainties, air mass factors, differences between coincident observations, quality control data selection standards, among other factors. Merging the data required difficult choices. The SO2 total emission from each source is then calculated by combining the overpass sample of the plume mass with ECMWF reanalysis wind information and loss rate estimates. All of these considerations are documented in the text and figures.

The introduction of two additional satellite data sources offers new challenges as described in the text. Inevitable differences require explanations that should lead to a better understanding of the measurement and retrieval methods.

We would like to thank the reviewer for his favorable comment.

**Specific Comments**

Because the retrieval algorithms derive slant column SO2 amounts an air mass factor is required to convert them to a quantitative geophysical measure, namely the vertical column SO2 amount. The differences between the AMF choices almost suggest that this is a dark art. The authors make clear that the height of the absorbing layer is a primary factor affecting the AMF, although other factors such as partial cloud corrections also play a role. They chose to use independently calculated site-specific AMFs, similar to the original catalogue methods. The procedure is documented and referenced.

The need for arbitrary corrections (Ie, 22%) is disturbing. It's surprising that the TROPOMI data production group uses SO2 absorption cross sections suitable for tropopause level air temperatures when a primary mission objective is to measure surface and low altitude plumes. Theys, et al. (2017) contend that this is "in principle accounted for in the AMFs". So why do we need a 22% correction?

It is not arbitrary. The correction reflected the difference in the SO2 cross-sections used in the OMI/OMPS vs. the TROPOMI COBRA retrievals. TROPOMI used an SO2 cross-section for a stratospheric temperature, where OMI/OMPS used one for surface temperatures. TROPOMI COBRA algorithm indeed corrected for this difference when AMFs were applied to calculate VCDs. However, while we used TROPOMI COBRA SCDs, we applied our own AMFs. Therefore, the correction was necessary. We added additional explanation of this to the text.

It is concerning that the OMI data are empirically corrected by 10% to force agreement with the original catalogue results. Please explain the rationale for this correction. Are more ground- truth estimations now available since the first catalogue? A discussion of the ground truth analysis would be appropriate especially since the original catalogue had empirical corrections to "ensure agreement with reliable at stack emissions measurements...".

We found that emissions calculated using Version 2 OMI/OMPS data are different from those, calculated from the previous version of OMI/OMPS data. There are many sources of a possible multiplicative bias in the estimated emissions. The original catalogue was validated against measured emissions from the US power plants and the calculated AMFs were adjusted to match the measured emissions. For version 2, we adjusted OMI/OMPS-based emissions to match emissions from the original catalogue. For this reason, at 10% correction was introduced. This was mentioned in the text, and we added more explanation.

The procedures for merging the three data sets are well documented. The difficulties of multiple-source regions and the effect of ground resolution differences are described. Lumping the former USSR countries together is no longer meaningful.

We changed "former USSR" to "Northern Eurasia" (Russia, Ukraine, Kazakhstan, and former USSR countries in Central Asia). To our best knowledge, no regulations to reduce power plant emissions were introduced. in these countries. This makes that region different from, for example, Estonia where emissions have been reduced substantially since Estonia joined the EU.

The reference list is extensive.

Technical corrections

Line 183 sp fund => find

Corrected

Line 203 sp enshure => ensure

Corrected

Line 275 ? "....and OMPS data do not produce a reliable fit. This may explain why OMPS-

based emissions are biased low." This requires a bit of explanation.

We just wanted to state that OMPS estimates for small sources are less reliable due to high uncertainties (compared to OMI and TROPOMI). We corrected the text to make it clear.

Line 313. Repeat "that that"

Corrected

L. 321, 322, 331: "Figure 5" is incorrect. Figure 6?

Figure 5 should be Figure 6 and vice versa. Corrected.

L 386 – 7. "The estimated emissions are about 40 kt y-1 but it is likely the combined emissions

from the powership and the existing power plant in the area." Incomplete sentence.

L 431. increase => increased

We changed that sentence and corrected the error.

L 456 typo weighed = > weighted

Corrected.

L 475 typo utilises => utilizes

Corrected.

L 498 "Can L" should be "Can Li"

Corrected.

I suggest running a spell check on the document. I doubt that I have caught all the errors.

We corrected these and some other typos.

---

## Referee Report (RR1)

General comments

This paper describes a new catalog of annual emissions from anthropogenic and natural sulfur dioxide sources based on Fioletov et al.'s (2016) original 2005-2014 OMI-based catalog. It incorporates OMPS and TROPOMI data as they became available during this extension to 2021. It uses updated OMI/OMPS algorithm results and, with TROPOMI's higher ground resolution, has resulted in the detection of more sources. Rationalizing differences between the three instruments is not trivial, particularly for smaller sources or clustered sources, as described in the text.

This paper is an important contribution to the global identification and monitoring of sulfur dioxide emissions. The global changes over the 16-year period are particularly interesting, showing significant decreases in the major producers, China, the USA, and Europe, while other regions are unchanged or show lesser decreases. India is the outlier in showing an increase. Volcanic degassing has also decreased although not as rapidly as industrial sources. Maps of sources and the distribution by source type and region and their changes between 2005 and 2021 are valuable products.

The industrial power plants, smelters, and oil and gas sources and their changes over the years are described in some detail, demonstrating the power of the satellite monitoring capability. The discussion of new sources is particularly interesting, such as the detection of ship-borne power plants in several ports.

Specific comments

The authors have addressed the primary issues raised by the reviewers. For example, the replacement of the group "former USSR countries" with "northern Eurasia countries" is appropriate. Editorial corrections to some lesser issues fail to fix the problems and in one case even made them more confusing. In response to R#2's question about the 22% adjustment to TROPOMI data the "added additional explanation" does nothing to elucidate the matter. In fact, the original text is preferable. Even the references do not explain how the cross sections are adjusted for temperature differences. It appears that the 22% value comes from Theys, et al. (2016) Figure 6 value of the "SO2 SCD effect" for the 312-326 nm window at 293K. This paper may not be the best place to clear up this issue. If another paper is available, please cite it.

Technical corrections

Line 80. "the whereas newest" likely should read "whereas the newest"

Line 231. The response to R#2's question about a 10% correction contains a typo: "measures" should read "measured" emissions.

Line 145 – 6. "the calculated in this study TROPOMI SO$_2$ VCD values…", probably should read: "the TROPOMI SO$_2$ VCD values calculated in this study…."

---

## Author Response (AR2)

**General comments**

This paper describes a new catalog of annual emissions from anthropogenic and natural sulfur dioxide sources based on Fioletov et al.'s (2016) original 2005-2014 OMI-based catalog. It incorporates OMPS and TROPOMI data as they became available during this extension to 2021. It uses updated OMI/OMPS algorithm results and, with TROPOMI's higher ground resolution, has resulted in the detection of more sources. Rationalizing differences between the three instruments is not trivial, particularly for smaller sources or clustered sources, as described in the text.

This paper is an important contribution to the global identification and monitoring of sulfur dioxide emissions. The global changes over the 16-year period are particularly interesting, showing significant decreases in the major producers, China, the USA, and Europe, while other regions are unchanged or show lesser decreases. India is the outlier in showing an increase. Volcanic degassing has also decreased although not as rapidly as industrial sources. Maps of sources and the distribution by source type and region and their changes between 2005 and 2021 are valuable products.

The industrial power plants, smelters, and oil and gas sources and their changes over the years are described in some detail, demonstrating the power of the satellite monitoring capability. The discussion of new sources is particularly interesting, such as the detection of ship-borne power plants in several ports.

We would like to thank the reviewer for his favorable comment.

**Specific comments**

The authors have addressed the primary issues raised by the reviewers. For example, the replacement of the group "former USSR countries" with "northern Eurasia countries" is appropriate. Editorial corrections to some lesser issues fail to fix the problems and in one case even made them more confusing. In response to R#2's question about the 22% adjustment to TROPOMI data the "added additional explanation" does nothing to elucidate the matter. In fact, the original text is preferable. Even the references do not explain how the cross sections are adjusted for temperature differences. It appears that the 22% value comes from Theys, et al. (2016) Figure 6 value of the "SO2 SCD effect" for the 312-326 nm window at 293K. This paper may not be the best place to clear up this issue. If another paper is available, please cite it.

Indeed, the correction is based on Theys, et al. (2017), their Figure 6. We added a reference to that figure and removed the sentence about COBRA VCD to avoid potential confusion mentioned by the Reviewer. However, this is not a new adjustment. We only implemented the correction introduced and discussed in previous publications.

**Technical corrections**

Line 80. "the whereas newest" likely should read "whereas the newest"

Corrected

Line 231. The response to R#2's question about a 10% correction contains a typo: "measures" should read "measured" emissions.
Corrected

Line 145 – 6. "the calculated in this study TROPOMI SO2 VCD values…", probably should read: "the TROPOMI SO2 VCD values calculated in this study…."

Corrected